# Investigations into Ti-Based Metallic Alloys for Biomedical Purposes

Ildiko Peter

Department of Industrial Engineering and Management, Faculty of Engineering and Information Technology, University of Medicine, Pharmacy, Science and Technology "George Emil Palade" of Târgu Mureş, Str. N. Iorga nr. 1, 540139 Târgu-Mureş, Romania; ildiko.peter@umfst.ro

**Abstract:** In the present research paper, two systems based on Ti-Nb-Zr-Ta and Ti-Nb-Zr–Fe, containing non-toxic elements, are considered and investigated. The first aim of the paper is to enlarge up-to-date developed β-type Ti alloys, analyzing three different compositions, Ti-10Nb-10Zr-5Ta, Ti-20Nb-20Zr-4Ta and Ti-29.3Nb-13.6Zr-1.9Fe, in order to assess their further employment in biomedical applications. To achieve this, structural, microstructural, compositional and mechanical investigations were performed as part of this study. Based on the results obtained, the alloy with the highest Nb content seems to be the most appropriate candidate for advanced biomedical applications and, in particular, for bone substitution.

**Keywords:** metallic biomaterials; Ti-based alloys; microstructure; mechanical properties

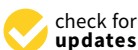



## 1. Introduction

Recently, research in the field of metallic biomaterials has focused on the development of new, improved and safe metallic biomaterials, with high biological and mechanical biocompatibility, in order to avoid any harmful effects from elements such as V and Al, which are usually present in a commercially available alloy (TiAl6V4 alloy) [1–3]. Even if their surface is functionalized, with different layers having high biocompatibility and bioactivity, there is a risk of hazardous effects on the human body caused by the toxic elements contained, if, over time, corrosion due to biological fluid ends up transferring ions from the implant material.

For the sake of completeness, it must be mentioned that some authors do not agree that the Ti6Al4V grade 23 displays non-toxic behavior, because of its V and Al contents [4–7]. Possible Fe toxicity has also been evidenced [8], since enhanced iron assimilation might be responsible for an excess of Fe content in different human organs (liver, pancreas, heart, etc.), which causes harm to tissues, weakening their correct functions, as well as leading to the development of various serious diseases.

The design of new biomaterials also requires a consideration of the need for their mechanical biocompatibility, which is as important as biological compatibility, consisting of high mechanical strength, wear resistance, fatigue strength and low elastic modulus, close to that of bone, especially when the implant is expected to bear load. The main areas of interest in the current research on biomaterials are focused on the development of biomaterials with composite structures and multilayer or surface-functionalized structures with micro- and nano-layers, in order to achieve good biocompatibility, acceleration of the osseointegration of the implant in the patient's body and prevention of post-implant infections, as well as direct surface manipulation [9–12].

For this purpose, there are several studies that estimate the relationship between the chemical composition and the microstructure of the surface, the physical-mechanical properties of biomaterials, the substrate and the coating material, and the properties of the human environment [13–16].

On the other hand, there has been an extensive effort to develop new eco-friendly technologies for obtaining biomaterials. Such methods are developed in such a way as to improve quality of life without polluting the environment and for workplaces where production is carried out. Additionally, eco-friendly implants guarantee economic efficiency and reproducibility, offering enhanced characteristics and performance compared to traditionally obtained materials [16–18]. Even though metallic biomaterials have been employed since the late 1960, they currently receive a great deal of attention; recently, there has been a huge inclination toward the use of metallic alloys, which have increased in importance through their extensive employment as biomaterials, especially in medical implantology. In this context, Ti and its alloys have received much more attention in bone surgery due to their excellent biocompatibility. Their biocompatibility is mainly attributed to specific physical-mechanical properties, such as their weight, mechanical resistance, modulus of elasticity, fatigue resistance and wear resistance in load, which are better adapted to those of human bone compared to other metal implant materials. Additionally, their surface is permanently covered by a native oxide film with a thickness of a few nanometers, which is responsible for excellent chemical inertia, corrosion resistance and bio-inert behavior in vivo; the passivation ability and even the biocompatibility of Ti and its alloys are the result of the chemical stability and structure of the $TiO_2$ film. A major trend in innovative Ti-based biomaterials is the development of Ti alloys made of non-toxic alloys for inactive implantable medical devices. Elements such as Nb, Ta and Zr, which have been shown by biological and clinical tests to be non-toxic and non-allergenic based on reported data on cell viability, corrosion resistance in biological fluids and compatibility with human tissue, have been chosen to be part of the composition based on Ti. Nb, Ta, and Zr have been shown to be the healthiest elements in alloy compositions. In addition, Fe, Mo and Sn have been accepted as elements without significant toxic effects on the growth and proliferation of biological tissue cells [19–23].

According to some investigations [24–26], stainless steel, Ti and Co-Cr metallic alloys, as the first generation of materials used for implant development, show excellent corrosion resistance and stability within the body. However, in recent years, there has been a significant development in the knowledge related to metals and their alloys employed for implant development since the appearance of biodegradable metals, such as Mg, Zn and Fe, as reported in [27]. These biodegradable metallic materials can definitely degrade in the body as soon as the scheduled activity is accomplished. The main challenges for Fe are fundamentally related to its outstanding mechanical characteristics and good biocompatibility which, in combination with its good formability, allow the development of thinner and more complex geometric parts [28]. According to the results of experimental research, carried out in a living organism, Fe offers reduced thrombogenicity and insignificant inflammatory reactions. However, research to explore Fe's toxicity is an on-going activity; some results are reported in [29,30].

In [31], the authors report that the main reason for the delay of the development of biodegradable Fe is its moderately low corrosion speed: no toxicity connected to a pure Fe stent was detected during the course of one year of observation, which was also confirmed in [32].

In the present research paper, two systems based on Ti-Nb-Zr-Ta and Ti-Nb-Zr-Fe, made up of non-toxic elements, were chosen, in order to contribute to the enrichment of up-to-date and developed β-type Ti alloys for biomedical applications. Structural, compositional and mechanical investigations were carried out in order to further propose the most suitable multifunctional metallic alloy for medical device applications.

## 2. Materials and Methods

Considering the influence of the alloying elements on the physical-mechanical properties and on the corrosion resistance, as well as the phase diagrams of the constituents of the chosen systems, the composition of the bio-alloys was established, in % of mass, as follows:

1. Ti-10Nb-10Zr-5Ta.
2. Ti-20Nb-20Zr-4Ta.

3. Ti-29.3Nb-13.6Zr-1.9Fe.

On the one hand, during the author's research into different metallic alloys for biomedical purposes (Ti-based alloys and modified ones, CoCr alloys, etc.), as reported in [33–35] and considering the outcomes of such investigations, the compositions proposed in this manuscript are those reported above. On the other hand, recent developments in research on metallic biomaterials and the requirements for Ti alloy biomaterials (high biocompatibility, no adverse effects on human tissue, high corrosion resistance in biological fluids, high mechanical strength, fatigue strength, low elasticity, low density, good wear resistance) were taken into account; based on these developments and requirements, the following two element systems were selected for the alloys investigated in this paper: Ti-Nb-Zr-Ta and Ti-Nb-Zr-Fe. Considering the influence of the alloying elements on the physical-mechanical properties and the corrosion resistance, as well as the phase diagrams of the constituents of the chosen systems, the composition of the bio-alloys was established, in% by mass, as follows: (1) Ti-10Nb-10Zr-5Ta; (2) Ti-20Nb-20Zr-4Ta; (3) Ti-29.3Nb-13.6Zr-1.9Fe [36–38].

The chemical composition of the alloys investigated is reported in Table 1. Compared with traditionally employed Ti-based alloys, including Ti6Al4V, the proposed new compositions, identified on the basis of the data reported in some previous papers [34–38], by changing the ratio of the elements and substituting some of them with others, such as Nb, Ta, Zr, Mo, etc., in different ratios, can offer some benefits: good corrosion resistance, non-toxicity of the elements of which the alloys are comprised, good mechanical resistance and reduction of the Young modulus, which is important for preventing excessive tension in the bone when it is in contact with the alloy. Additionally, compared to the traditional techniques (melting and casting) the considered innovative technology, namely the induction heating system used to obtain the bio-alloy, leads to the reduction of the excessive macro-segregation within the structure, which usually occurs during casting, as is also demonstrated in [39,40].

**Table 1.** Chemical composition of the alloys investigated.

| Alloys | | | Composition | | | | | Total Charge |
|---|---|---|---|---|---|---|---|---|
| | | | Ti | Nb | Zr | Ta | Fe | |
| 1 | Ti10Nb10Zr5Ta | (% mass) | 75.0 | 10.0 | 10.0 | 5.0 | – | |
| | | (g) | 112.5 | 15.0 | 15.0 | 7.5 | – | 150.0 |
| 2 | Ti20Nb20Zr4Ta | (% mass) | 56.0 | 20.0 | 20.0 | 4.0 | – | |
| | | (g) | 84.0 | 30.0 | 30.0 | 6.0 | – | 150.0 |
| 3 | Ti29.3Nb13,6Zr1.9Fe | (% mass) | 55.2 | 29.3 | 16.6 | – | 1.9 | |
| | | (g) | 82.8 | 43.9 | 20.4 | – | 2.9 | 150.0 |

The investigated alloys were synthesized using a melting furnace with cold crucible (in levitation) FIVES CELES, (CELES MP-25,Lotenbach, France)with a nominal power of 25 kW and a melting capacity of 30 cm$^3$, starting from the elemental components.

Since it is proposed that the alloys should be used in medical applications, it is necessary to rigorously observe the quality of the metallic materials used in the synthesis of these materials in order to obtain high quality alloys. This aspect was carefully controlled, since the level of purity of the raw materials influences the content of the impurities in the final alloy, including the gaseous ones (nitrogen, hydrogen), which are very strictly limited.

To obtain TiNbZrTa and TiNbZrFe alloys in the cold crucible melting furnace, the following starting materials were employed:

- metallic Ti with controlled oxygen content (Ti grade 3), with a composition according to DIN 3.7055, containing 0.30% Fe, 0.05% N$_2$, 0.25% O$_2$, max. 0.013% H$_2$, 0.10% C, balance Ti, provided by ZIROM Giurgiu, Romania.
- metallic Nb, 99.81% containing: 0.005% Fe; 0.005% Si; 0.010% Mo; 0.010% W; 0.002% Ti; 0.002% Cr; 0.1% Ta; 0.005% Ni; 0.02% O$_2$; 0.02% C; 0.0015% H$_2$; 0.015% N$_2$; balance Nb (Sigma Aldrich, Burlington, MA, USA).

- metallic Zr, 99.6% with the following composition: 0.01% Fe; 0.035% Si; 0.03% Mo; 0.05% W; 0.01% Ti; 0.02% Ni; 0.02% $O_2$; 0.01% C; 0.0015% $H_2$; 0.01% $N_2$; 0.2% Nb; balance Zr, provided by ZIROM Giurgiu, Romania.
- metallic Ta, 99.59% with the following composition: 0.01% Fe; 0.05% Si; 0.02% Mo; 0.05% W; 0.01% Ti; 0.01% Ni; 0.03% $O_2$; 0.01% C; 0.0015% $H_2$; 0.01% $N_2$; 0.2% Nb; balance Ta (Alfa Aesar Chemicals, Ward Hill, MA, USA).
- Fe, with: 0.015% C; 0.01% Si; 0.02% Mn; 0.02% S; 0.01% P; 0.015% $O_2$; balance Fe (Sigma Aldrich, Burlington, MA, USA).

The synthesis of the alloys in the cold crucible furnace included the following main phases and operations: the preparation of the raw materials by cutting them to appropriate dimensions; ultrasonic cleaning; degreasing with volatile organic solvents (e.g., acetone); the weighing/dosing of the raw materials according to the batch calculation; loading the raw materials into the oven crucible; vacuuming the installation and creating a controlled atmosphere (Ar) in the melting chamber; melting the charge by adjusting the electric power; casting; cooling and the evacuation of the ingot from the ingot mold; loading the ingot into the crucible of the oven, for re-melting; vacuuming the installation for the elimination of residual gases from the melting chamber, followed by the establishment of a controlled atmosphere (Ar) for melting; melting the charge by adjusting the electric power; casting the final ingot; cooling and evacuation of the first melting ingot from the ingot mold; obtaining the final ingot by turning.

Below, the technological parameters for the synthesis of alloys in the cold crucible furnace are reported:

1. ingot diameter: 20 mm; -vacuum (primary and secondary).
   - primary vine: $4 \times 10^{-2}$ torr.
   - secondary vine: $9 \times 10^{-4}$ torr.
   - inert working atmosphere (argon): $-0.3$ bar.
2. electrical parameters of the levitation melting furnace.
   - power: 23–24 kW.
   - frequency: 105–110 Hz.
3. cooling of the vacuum system, generator and melting module (crucible/ingot mold): continuous.

Three ingots were obtained, one from each type of alloy, which, after turning, featured a diameter of approximately 18.5 mm and lengths of 61 mm (alloy 1), 60 mm (alloy 2) and 54 mm, respectively, as shown in Figure 1.

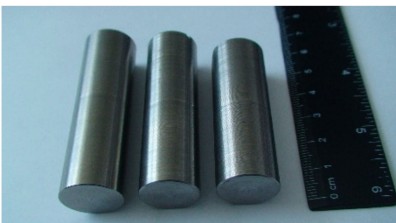

**Figure 1.** Photograph of the ingots obtained.

The chemical composition of the cast alloys was determined using an X-ray fluorescence spectrometer (XEPOS 03 type, Spectro Analytical Instruments GmbH, Kleve, Germany) with a specialized program for the analysis of the metallic materials. For better accuracy, the measurements were performed under vacuum. The elemental composition and the distribution of the alloying elements were analyzed by SEM-EDS and SEM-BSE electron microscopy (SEM, TESCAN VEGA II-XMU SEM microscope type, Tescan Analytics, France). The cutting of the ingots was carried out using a precision cutting machine (MICRACUT 201 diamond disc, Metcon Technology, Monaca, PA, USA), obtaining discs with a thickness of 3.4 mm. The obtained samples were hot-embedded in Buehler

Konducto Met resin (ECOPRESS 100 embedding machine, Micro Optics, SIngapore) and subsequently sanded and polished. Immediately after finishing, the samples were etched (5% HF + 15% HNO$_3$ + 80% H$_2$O) in order to monitor their microstructure. The etching solution was used fresh, immediately after preparation. The samples were immersed for about 10 s, then they were washed in water and dried. The metallographic analysis was performed immediately after the chemical attack of the samples. The main microstructural characteristics of the alloys in cast state were highlighted by optical microscopy (OM, Olympus BX51M microscope equipped with ATM Inspect software, Microscope Systems Limited, Glasgow, UK) and a SEM. The quantitative analysis of the constituent phases was performed with OM equipped with ATM Inspect Phase Analysis software. The composition of the phases was revealed by X-ray analysis technique (XRD, PANalytical X'Pert PRO MRD, Philip PW 3710 diffractometer, Tokyo, Japan). The size and distribution of the grains were monitored with a SEM with EBSD Bruker Quantax eFlash 1000 tool. The following equipment was used in the preparation of the cast alloy strips and their thermo-mechanical processing:

- MICRACUT 201 precision cutting machine, with diamond disc, with variable cutting speed, table with automatic movement on the X-axis with digital positioning, power 1 kW.
- Rolling mill (Mario di Maio, LQR120AS, laboratory mill) with a power of 4 kW, with variable working speed.
- Heat treatment furnace (Nabertherm High Temperature Furnace HTC 08/16/P330 LC080K6SN, Germany) with a maximum heating temperature of 1600 °C; volume 15 l, accuracy higher than 1 °C, with the possibility of programming the treatment diagram (programming was performed in steps of 1 °C or 1 min).

The thermo-mechanically processed samples were characterized from a mechanical point of view (Young modulus, ultimate tensile strength, yield strength = −0.2% proof stress, elongation to fracture) using a GATAN Micro Test 2000 module machine (Gatan GmbH, Munich, Germany) mounted in a SEM. The samples subjected to elongation that demonstrated the standard shape and dimensions were obtained using a wire EDM cutting machine.

For the thermo-mechanical processing of the alloys, the following steps were adopted:

- cold rolling, with the alloy samples in the cast state, cut to the appropriate thickness, deformed with a maximum degree of 10% at each pass; the total degree of deformation was specific to each type of alloy.
- recrystallization treatment, which was necessary to finish the structure of the alloys after rolling, with the following parameters: recrystallization temperature; recrystallization time (maintenance at the highest temperature); cooling in water.

The technological flow included the following main phases and operations:

- cutting the ingot in a cast state to obtain the alloy strips.
- cold rolling of the cast alloy strips.
- recrystallization of laminated strips.

With the help of diamond disc cutting machines, the blades with the thickness specified in Table 2 were cut from the ingots in the cast state.

**Table 2.** Some mechanical properties of the alloys.

| Alloys | Ultimate Tensile Strength $\sigma_{UTS}$ [MPa] | 0.2 Yield Strength $\sigma_{0.2}$ [MPa] | Elongation to Fracture $\varepsilon$ [%] | Elastic Modulus E [GPa] |
|---|---|---|---|---|
| Ti10Nb10Zr5Ta | 742 ± 10 | 397 ± 8 | 13.9 ± 0.2 | 44 ± 2 |
| Ti20Nb20Zr4Ta | 716 ± 10 | 335 ± 5 | 14.8 ± 0.1 | 42 ± 3 |
| Ti29.3Nb13.6Zr1.9Fe | 995 ± 3 | 570 ± 7 | 7.6 ± 0.3 | 46 ± 5 |
| Ti29.3Nb13.6Zr1.9Fe laminated | 1355 ± 5 | 690 ± 5 | 8.7 ± 0.2 | 51 ± 2 |
| Ti29.3Nb13.6Zr1.9Fe recrystallized for 5 min | 933 ± 1 | 824 ± 3 | 14.2 ± 0.2 | 65 ± 5 |
| Ti29.3Nb13.6Zr1.9Fe recrystallized for 15 min | 909 ± 4 | 786 ± 5 | 8.8 ± 0.1 | 63 ± 5 |
| Ti6Al4V [33,37] | 895–1300 | 830–1100 | 10 | 100–120 |

## 3. Results and Discussion

### 3.1. Chemical Characterization and Structural Analysis of the Cast Alloys

When establishing the appropriate melting process for the development of the alloys with the compositions indicated, the following aspects were taken into account, which imposed certain requirements on the development process:

- the elements in the alloys demonstrated great differences between their melting temperatures (from 1538 °C in the case of Fe to 1668 °C for Ti, 1855 °C for Zr, 2500 °C for Nb and about 3000 °C for Ta), which influenced the choice of synthesis process, because the temperature reached in the processing furnace needed to be high enough to obtain a homogeneous composition without un-melted metal after the solidification.
- the alloys contained elements with densities with different ratios: 1:1.5, 1:1.7; 1:2 and 1:4, which made it difficult to create a uniform composition, which then required an intensive homogenization system.
- the binary equilibrium diagrams of Ti with Nb, Zr, Ta and Fe highlighted their total liquid solubility with the formation of solid solutions.
- heavy reactivity of the main elements in the composition of the alloys as the temperature increased towards the gases (oxygen, nitrogen, hydrogen), starting from 250 °C, with the formation of compounds that excessively hardened the alloys, developing centers of chemical and structural non-uniformity that affected the resistance corrosion.

Considering all the aspects presented above, melting in a cold crucible furnace (in levitation) was employed for the alloy synthesis, an approach also used by other researchers [39,40] for the development of different compositions. Magnetic levitation melting in a cold crucible furnace (CCLM) is known as a clean melting method without contamination of molten material. This melting method offers various advantages, such as: metals with different melting points can be melted, including metals with very high melting temperatures; non-contamination of the materials melted in levitation inside the crucible; a very high process speed; good homogeneity of alloys due to electromagnetic forces that produce a strong melting effect; the possibility of pouring samples into desired shapes, within the limits allowed by the dimensions of the oven; high cooling speed of the cast ingot, which allows the obtainment of a structure with fine grains, which is necessary for efficient thermomechanical processing of the alloy; and melting can be performed in a vacuum or in a controlled atmosphere.

As far as possible, the elements of which the structure was comprised needed to be β-stabilizers (such as Nb, Zr, Ta) in order to develop β-type Ti alloys more easily. According to [41], Ti alloys can turn into complete β structures at room temperature; in this study a high level of Nb is proposed. Additionally, due to the fact that Ta is an expensive element [42], here its replacement with Fe is proposed. The compositions obtained, reported in Table 3, were very close to the nominal composition of the alloys. The analysis was carried out using XRF with a specialized program for the analysis of metallic materials and, for superior accuracy, the measurements were performed under vacuum. Three measurements were performed, in three different ingot positions: at the two ends of and at the middle of the ingots. This was in order evaluate the homogeneity and uniformity, as composition concerns, of the obtained alloys. There were insignificant differences in the results obtained, which is why it the standard deviation is not indicated.

**Table 3.** Chemical composition of the alloys in casted state.

| Alloys | Composition of the Alloys (% wt) | | | | |
|---|---|---|---|---|---|
| | Ti | Nb | Zr | Ta | Fe |
| Ti-10Nb-10Zr-5Ta | 74.89 | 10.06 | 9.94 | 5.01 | |
| Ti-20Nb-20Zr-4Ta | 55.93 | 20.01 | 19.92 | 4.03 | |
| Ti-29,3Nb-13.6Zr-1.,9Fe | 55.10 | 29.39 | 13.55 | | 1.91 |

At the moment of the turning of the casted ingots, a superior processability/workability was observed for the Ti29.3Nb13.6Zr1.9Fe alloy, compared to the other two compositions, which was principally attributable to the higher Nb content. As reported in [43,44], the addition of Nb to an Hf-containing Ni-Ti alloy contributes to the enhancement of the ductile property without considerably modifying other functional behaviors. As reported in [43], this fact is partially due to the development of a eutectic constituent, such as (semi-) coherent soft β-Nb, making wire drawing of the alloy possible, which is directly correlated to good workability.

The EDS spectrum and the chemical composition obtained after the analysis are illustrated in Figure 2, while Figure 3 reports the distribution of the alloying elements on the sample surface, demonstrating once again that the obtained compositions were comparable to the nominal composition of the alloys.

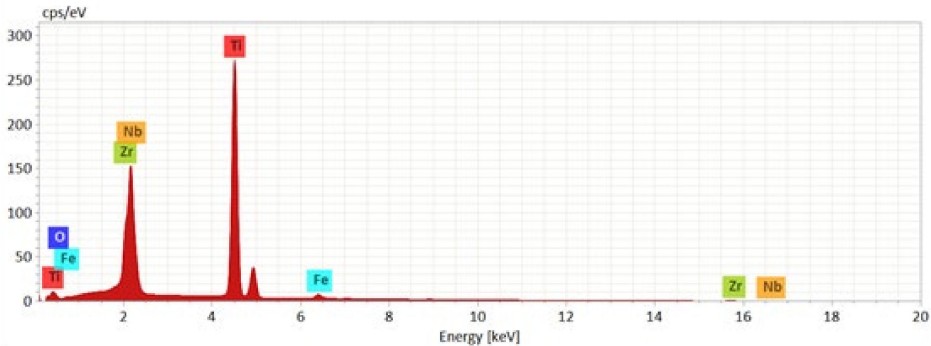

**Figure 2.** EDS spectrum for the sample Ti29.3Nb13.6Zr1.9Fe in cast state.

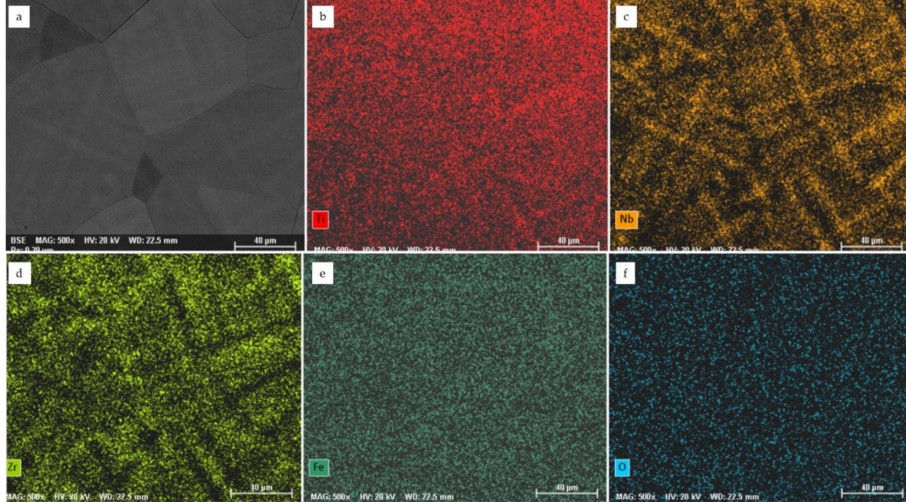

**Figure 3.** Distribution of alloying elements for the sample Ti29.3Nb13.6Zr1.9Fe in a cast state: (**a**) -SEM-BSE image of the analyzed field; (**b**) -Ti distribution; (**c**) -Nb distribution; (**d**) -Zr distribution; (**e**) -Fe distribution; (**f**) -O distribution.

The structure of the cast alloys is reported in Figure 4.

The Ti10Nb10Zr5Ta alloy demonstrated a lamellar appearance in its structure (Figure 4a,b), the α-Ti phase being the majority of the sample mass (Figure 5a,b). The microstructure of the cast Ti20Nb20Zr4Ta alloy demonstrated an appearance characteristic of the method by which the material was obtained, presenting a typical casting structure, with relatively coarse granulation, as reported in Figure 4c,d. Its structure consisted of large polygonal grains, the β-Ti phase being the majority of the sample mass (Figure 5c,d). As stated above, the aim was to have as much as possible of the β-Ti phase.

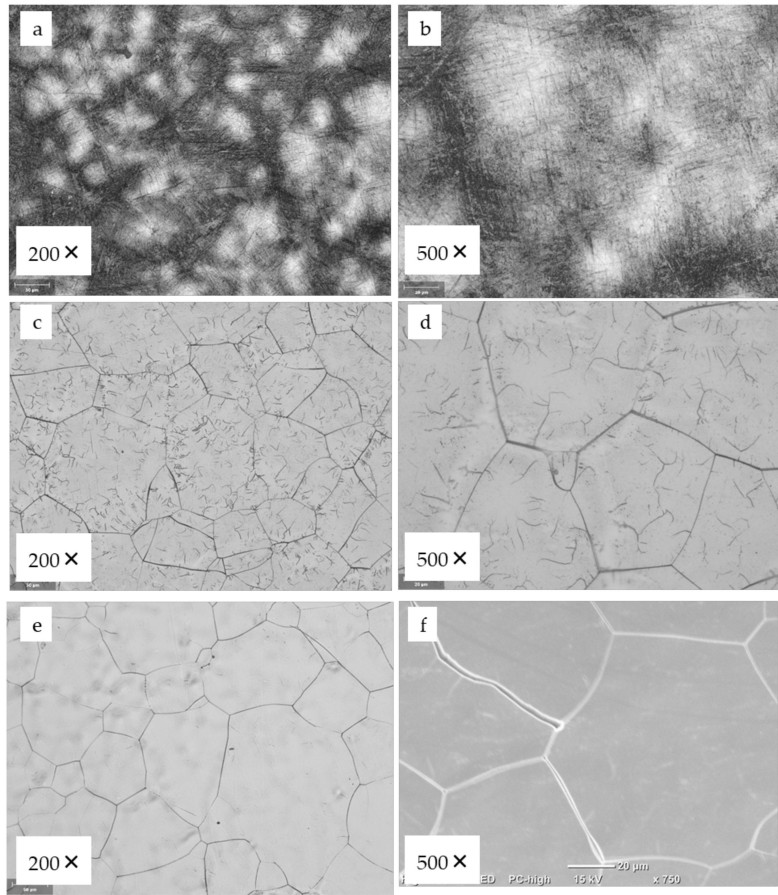

**Figure 4.** OM microstructure of the alloys: (**a**,**b**) Ti10Nb10Zr5Ta; (**c**,**d**) Ti20Nb20Zr4Ta; and (**e**,**f**) Ti29.3Nb13.6Zr1.9Fe in a cast state.

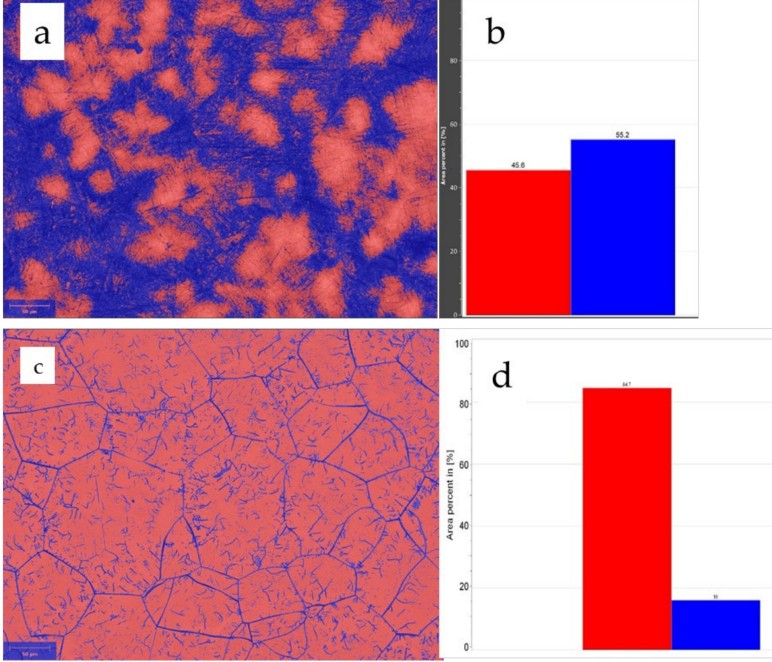

**Figure 5.** OM microstructure of the alloys: (**a**,**b**) Ti10Nb10Zr5Ta; (**c**,**d**) Ti20Nb20Zr4Ta in a cast state. Blue color indicates α-Ti phase while red color indicates β-Ti phase.

The Ti29.3Nb13.6Zr1.9Fe alloy also demonstrated a casting structure with large equiaxed grains (Figure 4e,f), consisting of the β-Ti phase, as was expected given the high concentration of β-stabilizing elements. The grains appeared uniformly in the structure, with an average size of 84.2 μm; 87 grains appeared on the analyzed surface (Figure 6).

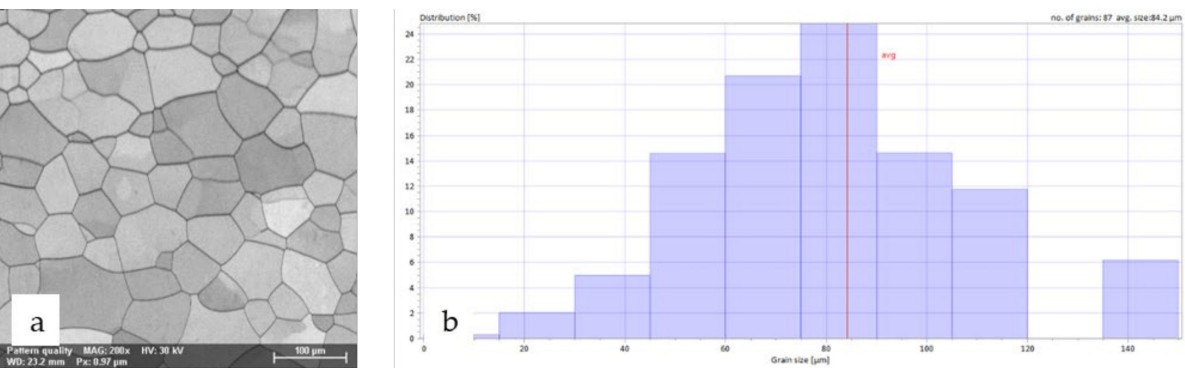

**Figure 6.** SEM microstructure (**a**) and the distribution of grain size (**b**) of the Ti29.3Nb13.6Zr1.9Fe alloy in as cast state.

### 3.2. Design of the Method of Thermo-Mechanical Processing of Alloys

Thermo-mechanical processing was considered in order to achieve a preferred combination of properties: the rough structure of the α or β grains needed to be converted into a structure made up of fine, uniform, equiaxed grains. Given the high content of Nb and Ta, elements that contributed to the major increase in the deformability of the alloy, but also the need to obtain a suitable structure (whose influence in terms of its mechanical properties and corrosion resistance was decisive), which satisfied the requirements imposed by the application targeted in this research, thermo-mechanical processing was performed. The thermo-mechanical processing parameters for the three samples were those reported in Table 2. For the Fe-containing alloy sample, which looked to be the most promising in terms of its mechanical properties, two recrystallization treatments were performed, with holding times of 5 and 15 min, respectively, to evaluate the changes in the sample's structure after the introduction of the variable.

### 3.3. Structural Characterization of Alloys after Recrystallization Treatment

The structural characterization of the alloys was carried out in order to investigate their morphology and homogeneity and, further, to obtain some indication as to their workability, an important issue given that the material needed to be processed for obtaining the final shape of the medical devices. The obtained microstructure of thermo-mechanically processed alloys is shown in Figures 7 and 8.

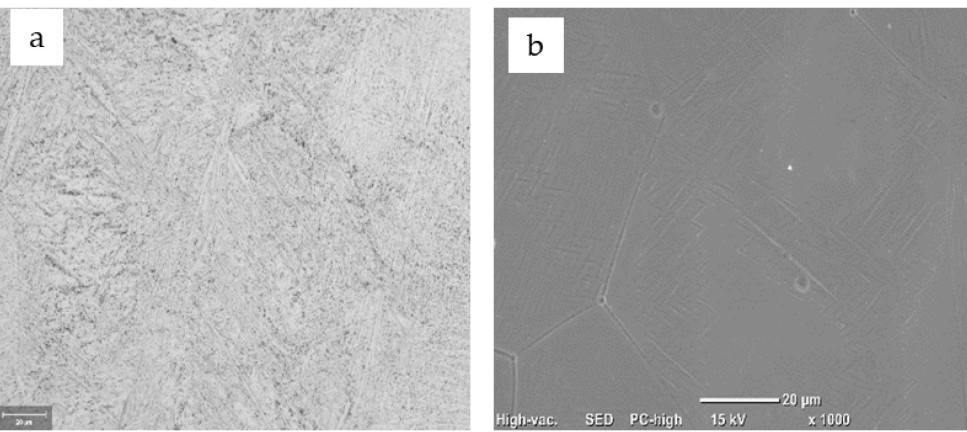

**Figure 7.** OM microstructure of the processed alloys: (**a**) Ti10Nb10Zr5Ta; and (**b**) Ti20Nb20Zr4Ta.

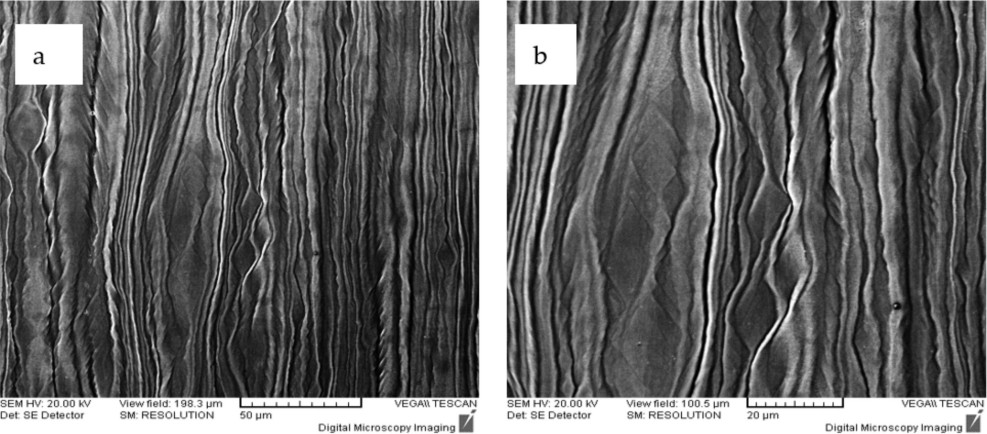

**Figure 8.** SEM microstructure of the Ti29.3Nb13.6Zr1.9Fe alloy processed at two (**a**,**b**) different magnifications.

The Ti10Nb10Zr5Ta and Ti20Nb20Zr4Ta alloys demonstrated a very low susceptibility cutting or processing. It was very difficult to sketch the different sections of the material, an indication that from application point of view, such alloys are hard to use. The established crystallographic textures were not homogeneous and there was the presence of defects (pores), which can negatively affect the overall performance of alloys. Once again, a higher Nb content contributed to a reduction in the hardness of the alloy as can be seen in Figure 9, the shear bands developed in the Ti29.3Nb13.6Zr1.9Fe alloy were not uniformly distributed in the structure and appeared curly. Commonly, such structures are characteristic of metals with bcc crystallografic structures, generated from swapped and heavily distorted grains because of the localization and growth of a quantity of shear strain on a slip plane. The alloy was strengthened by this process and the grain refinement occurred, after recrystallization. The fine grains produced led to a large quantity of grain boundaries.

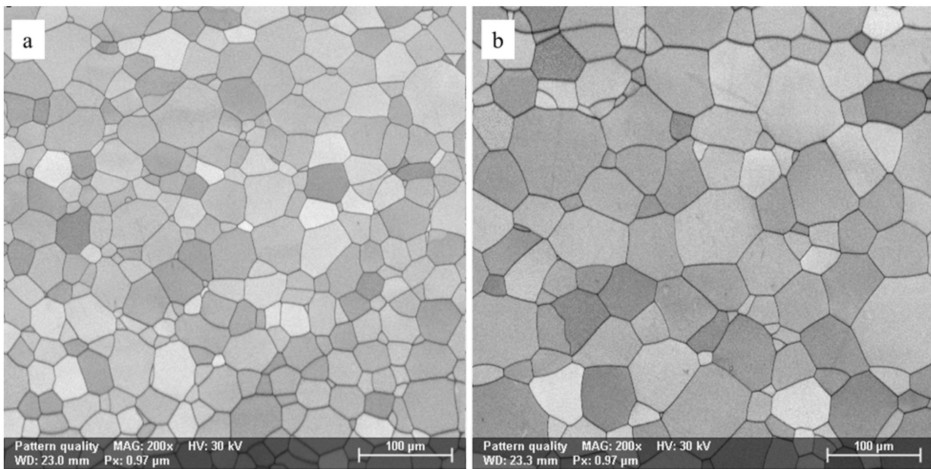

**Figure 9.** OM microstructure of the Ti29.3Nb13.6Zr1.9Fe alloy recrystallized after 5 min (**a**) and after 15 min (**b**).

Recrystallization was performed, using different times, in order to convert the microstructure and to control the grain arrangement in the material. This aspect is very important to guarantee a higher ductility in the material, even if a decrease in the strength and the hardness of the alloy is expected. After recrystallization, the grains turned into an equiaxed shape (Figure 9). The microstructure of the alloys after recrystallization was refined with defect-free grains determining improved properties compared to the coarse structure developed in the cast state. As the duration of the recrystallization increased from 5 min to 15 min, the grain size increased. The morphology and the distribution

of the grains for the recrystallized alloy (5, 15 min) are reported in Figure 10, while the number and the size of grains on the analyzed surface are reported in Table 4. The grain size also had a significant effect on the corrosion resistance, which increased as the grain size decreased [8]. However, at the same time, the development of different textures and the creation of internal stress can also affect the corrosion resistance and, according to one study [44], there are some unanswered questions as to the real relationship between grain size and corrosion resistance. From the analysis carried out in this study, it appears that recrystallization for 5 min is adequate for achieving the desired microstructural features; this also guarantees acceptable mechanical performances, as reported below.

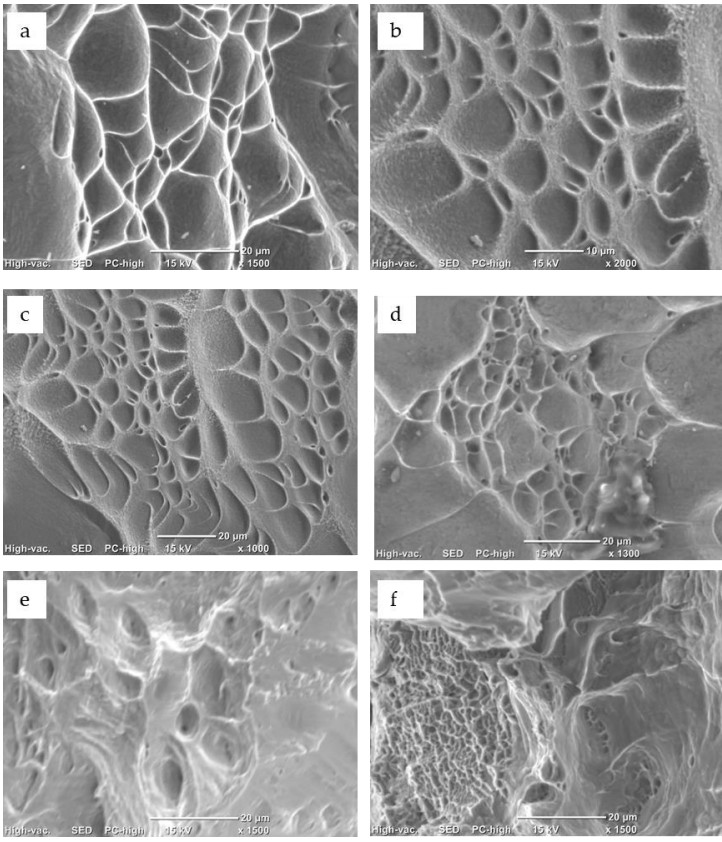

**Figure 10.** SEM fracture surface of the alloys processed: (**a**) Ti10Nb10Zr5Ta; (**b**) Ti20Nb20Zr4Ta; (**c**) Ti29.3Nb13.6Zr1.9Fe; (**d**) Ti29,3Nb13,6Zr1,9Fe laminated; (**e**)Ti29.3Nb13.6Zr1.9Fe recrystallized for 5 min; and (**f**) Ti29.3Nb13.6Zr1.9Fe recrystallized for 15 min.

**Table 4.** Process parameters for the thermo-mechanical treatment of the alloys.

| Alloy | Initial Thickness of the Strips [mm] | Rolling Speed [m/min] | Degree of Deformation at One Pass [%] | Total Degree of Deformation [%] | Final Thickness of the Strips [mm] | Recrystallization Temperature [°C] | Holding Time at the Maximum Temperature [min] | Cooling Medium |
|---|---|---|---|---|---|---|---|---|
| Ti10Nb10Zr5Ta | 4 | 3 | 10 | 50 | 2 | 920 | 15 | Water |
| Ti20Nb20Zr4Ta | 4 | 3 | 10 | 50 | 2 | 920 | 15 | Water |
| Ti29.3Nb13.6Zr1.9Fe | 6.5 | 3 | About 10 | 87 | 0.84 | 950 | 5 and 15 | Water |

### 3.4. Mechanical Characterization of Processed Alloys

The mechanical properties of the analyzed alloys are reported in Table 5. It is possible to observe that there were some differences between their performances. The Fe-free alloys demonstrated lower values for UTS and $\sigma_{0.2}$ compared to the third composition and to the reference Ti6Al4V alloy. The elastic modulus was close to the elastic modulus of the bone [44]; however, its workability, demonstrated during its processing for the sample preparation, can compromise its use in such applications. The UTS of the Fe-containing

alloy increased in all states and the values were in the range of the values reported for the reference alloy [44]. The morphology and the size of the grain directly controlled the mechanical resistance of the alloy. After rolling, the alloy was strengthened as a result of the cold plastic deformation, which generated dislocations, simplifying the sliding of the crystalline planes within the structure and leading to the highest UTS value. Softening occurred after recrystallization. In addition, compared to the biocompatible Ti6Al4V alloy, the advantage of all the experimental alloys consists in their constitutive elements, which are biocompatible and do not produce toxic or allergic effects along with their sufficient mechanical properties. All the values can be considered acceptable, and they are close to the properties of other biocompatible alloys [44]. The elastic modulus is about half of that of the Ti6Al4V alloy, an important aspect when the alloy is employed for repairing fractured bone, or replacing it. In the case of a material which shows a Young modulus comparable to that of bone, a more uniform stress distribution can be achieved and the transfer of the stress through the device and bone interface decreases the mechanically induced resorption of the bone.

**Table 5.** The number of grains and their size distribution in the Ti29.3Nb13.6Zr1.9Fe alloy.

| Alloy | Time of the Recrystallization [min] | Number of the Gains [nr.] | The Average Size of the Grains [μm] |
|---|---|---|---|
| Ti29.3Nb13.6Zr1,9Fe | 5 | 375 | 41.8 |
| | 15 | 154 | 64.5 |

Following the mechanical characterization, the fracture surfaces were analyzed. As reported in Figure 10, the predominant mechanism of the fracture consisted of a mixture of plastic flow and fracture (due to increasing ductile voids), both of which are characteristic of ductile behavior.

Fractures arose after a great plastic deformation, with no fibrous morphologies or cracks. In some cases, there were some voids and regions with marks of cavity combination, demonstrating evident ductility.

## 4. Conclusions

In this study, two systems based on Ti-Nb-Zr-Ta and Ti-Nb-Zr-Fe, using non-toxic elements, were prepared and characterized. The substitution of Ta with Fe was proposed, since it is more convenient from an economic point of view.

On the one hand, the aim was to investigate the obtained materials and, on the basis of the obtained results, to propose the most suitable metallic alloy for bone replacement, with enriched properties. On the other hand, the intention was also to enrich up-to-date developed β-type Ti alloys for biomedical applications.

Magnetic levitation melting in a cold crucible furnace was shown to be useful for the production of defect-free, homogenous alloys with no segregation in all the three alloy compositions.

Thermo-mechanical processing and recrystallization were performed to obtain a good combination of properties. The mechanical characterization demonstrated that the investigated compositions contain appropriate mechanical properties, which are in line with the characteristics of other Ti alloys commonly employed for similar applications. However, the lower elastic modulus obtained was about half that of the Ti6Al4V alloy, an important aspect if the alloy is employed for repairing fractured bone or substituting it. A compatible elastic modulus with that of bone can survive reasonably higher loads, avoiding the "stress shielding" commonly present in some currently used implant materials.

From the application point of view, one also has to consider the workability aspects. At the moment of the turning of the casted ingots, superior processability was observed for the Ti29.3Nb13.6Zr1.9Fe alloy compared to the other two compositions. This aspect is encouraging, since easier workability is associated to higher productivity, saving time and energy. The Ti29.3Nb13.6Zr1.9Fe alloy was selected as a more appropriate candidate for medical device applications. Further research will be performed (i) in order to verify

the non-release of simulated physiological fluids of toxic elements, with a particular focus on the potential hypothetical toxicity of Fe as a component of the tested alloys, taking into account investigations performed by other researchers; and (ii) the possibility of the further use of the functionalization of the Ti29.3Nb13.6Zr1.9Fe alloy in applications such as (1) smart and multifunctional medical devices and/or (2) implants coated by a metallic pattern and generating an innovative functional device.

**Funding:** This research was funded by a grant from the Romanian Ministry of Education and Research, CNCS-UEFISCDI, project number PN-III-P4-ID-PCE-2020-0404, within PNCDI III.

**Data Availability Statement:** Not applicable.

**Conflicts of Interest:** The author declares no conflict of interest.

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
