# Peer review of "Investigations into Ti-Based Metallic Alloys for Biomedical Purposes"

_metals, doi:10.3390/met11101626_

Round 1

Reviewer 1 Report

The problem considered at work is very important and undertaken by many scientists. The article deals with the new titanium alloys for biomedical applications. The problem is current and very significant in light of the continuous increase in the modification and use of new materials. Experimental tests, i.e. chemical analysis, microscopic, physicochemical analysis were performed. The manuscript is interesting and quite well written. However, there are several points that I would like to address:

  1. It is not clear on what basis the compositions of alloys were established (lines 86-92). Literature data or own results are recommended for confirmation.
  2. Not enough data of used powders (i.e. manufacturer, symbols, etc).
  3. Line 275 – please explain what the question marks mean.
  4. Line 338- please correct the description “and σ2 compared”
  5. Table 2 – the descriptions of columns are not clearly visible.
  6. Non-uniform writing of titanium alloy symbol in the manuscript.
  7. A deeper discussion of obtained data in comparison to literature data and other types of Ti alloys is highly recommended.
  8. Conclusion section – it is written that the goal of work was to obtain biocompatible alloys, however, there is no biological data for confirmation that obtained alloys are biocompatible.
  9. Literature list- non-uniform description of publications; position no 22 –please expand the description.

Author Response

Dear Reviewer,

First of all, I would like to thank you for your time dedicating to review my manuscript and for the recommendations

As from your letter, I have considered the objections, and modified the manuscript accordingly. Moreover, I have answered point wisely to the objections in the present letter.

In the present letter, my answers are reported in blue colour, while in the updated version of the manuscript the changes are highlighted, hence visible.

Best regards

Ildiko Peter

Reviewer 2 Report

Dear Authors,

The manuscript focuses on two systems based on Ti - Nb - Zr - Ta and Ti - Nb - Zr 71 - Fe using non-toxic elements were investigated for biomedical applications. Theses alloys are promising to be high biocompatible, high corrosive resistive to the biology environment and to provide high mechanical properties.

The content of the manuscript is relevant to the scope of the journal. However, the structure of the paper does not fit a research paper, some experiments show insufficient data, some data does not fit the goal, and some data looks as only added for the paper and not really obtained.

Here is my specific comments:

  1. Data on clinical tests using non-toxic Nb, Ta, Zr in lines 64–66 must be strengthened by relevant references.
  2. Please provide the manufacturer of all initial materials in the work.
  3. The fabrication method is briefly described however, I strongly recommend to made it more informative. E.g., you have to add temperatures, electrical parameters, times durations, cooling rates, etc.
  4. Please provide more informative parameters of the XRD analysis: type of radiation, geometry, electrical parameters, scanning rate.
  5. What the standard shape and dimensions are? (Line 186). Please add the parameters.
  6. Several flow issues should be solved. Results and Discussion part was started form some points which style is more fitted for the conclusions. Here, you have to describe your observation and to discuss them. Please re-write along the paper.
  7. “technology used also by other researchers for the development of different compositions” (lines 209–210). Please add relevant references.
  8. Table on page 5 must be changed to Table 2 and not Table 1. Is this XRF data? Please specify. How many measurements have been done? Please add standard deviation.
  9. Figures 5c and 5d contain very unclear and small numbers, please improve the quality and number appearance in the image.
  10. XRD plot in Figure 6 is totally confusing. How possible to get the only phase of Ti if your material contains toughly 30% of Nb and 13% of Zr. Why were these components not detected?
  11. What question marks in Line 275 for?
  12. All operations in Lines 276–285 must be moved to Materials and Methods section.
  13. Figure 10 is not a microstructure; it is an imitation of the structure!
  14. In Table 4 you have the same standard deviation in each column, how possible?
  15. Conclusions must be re-written. Current paper draft has some explanations and work aims; however, conclusions MUST CONCLUDE what did you get!

Author Response

(The authors gave the same response as above.)

Reviewer 3 Report

The reviewed article concerns the research of three alloys from the Ti - Nb - Zr - Ta and Ti - Nb - Zr - Fe systems. The assumption made at the basis for these studies is the statement that in the commonly accepted and used alloy for medical purposes Ti6Al4V grade 23, it is doubtful whether the V and Al contained in it have no toxic effect on the human body. In my opinion, this statement, established in the world's literature, is based on false assumptions, which have their source in the original work published several dozen years ago, and which is the result of commissioned research as part of a dispute between industrial lobbies which introduced, respectively, but competitively, Nb or V to micro-alloyed steels and consequently to alloys of other metals, e.g. titanium. The reliability of these reports is questionable, and the resulting information has been disseminated for decades. The author repeats this view, perhaps not being aware of the sources of this information. Although it is not of fundamental importance for the research carried out, it would be worth denying this untrue or at least doubtful information in the introduction to the reviewed article. It seems that it can be taken from the source research in the work of J. Biomed. Mater. Res. Part A 2012, 100, 768–775.https: //doi.org/10.1002/jbm.a.34006, as well as discussions on, among others in the work Processes 2020, 8, 664; doi: 10.3390 / pr8060664 and Introductory Chapter to Biomaterials in Regenerative Medicine Intech doi: 10.5772 / intechopen.73094. I think that it is easy to supplement the text with the relevant fragment, objectifying the one-sided and, in my opinion, the untrue view presented there, especially about the unfavourable role of vanadium in titanium alloys. Since three alloys Ti10Nb10Zr5Ta and Ti20Nb20Zr4Ta and Ti29.3Nb13.6zr1.9Fe were tested, and the analysis and the conclusions drawn indicate that the iron-containing alloy is the most advantageous of them, the lack of explanation of the mentioned problem opens a discussion about the possible iron toxicity, in my opinion completely redundant. However, going back to Genes & Nutrition Vol. 1, No. 1, pp. 25-40, 2006 ISSN 1555-8932 we find, however, that haemochromatosis is one of the most common congenital diseases (1 in 200 people in Europe), although there may also be secondary haemochromatosis. Increased iron absorption leads to excessive iron levels in the liver, pancreas, heart, damage to the structure and impairment of organ functions, development of cirrhosis and liver cancer, heart disease, and diabetes, as it is characterized by excessive iron deposition in tissues and organs. The author does not mention this (at least hypothetical) risk in the event of using the alloy selected by her. Perhaps it would also be worth considering this thread in the introduction, and perhaps even more carefully formulating the final conclusion. The aim of this article was to investigate the obtained materials and, on the basis of the obtained results, to propose the most suitable biocompatible alloy for bone replacement with enhanced properties. The proposed alloy can then be thermo-mechanically treated. On the other hand, the author's intention was also to enrich the offer of previously developed β-type titanium alloys for biomedical applications. The criterion for selecting an iron-containing alloy was its better machinability compared to the other two investigated alloys, which do not contain this element. Perhaps, however, taking into account the aforementioned aspect of the potential hypothetical toxicity of iron as a component of the tested alloys will result in the verification of the final and firm conclusion.
I believe that the article is valuable and deserves to be published as-is. However, I would suggest to the author to consider the above-mentioned corrections, without doubts being debatable, but I have the impression that with a very small amount of work, they can greatly improve the real value of the valuable work presented by her. In this case, I will write "minor changes", though I don't insist on it. I believe that the article should be published in the Special Issue on "Recent Biomedical Materials" in the Metals MDPI journal.

Author Response

(The authors gave the same response as above.)

Reviewer 4 Report

The author's purpose of the investigation is very interesting, also for scientists from related research fields. I would recommend the suggestions described below:

  • Abstract should be quantitative as possible for rapid comparison with others studies, referring quantitative values for the alloys studied. After reading the paper some info is missing in the abs. The abs should be a mirror of the paper and not a kind of intro, aims or approaches. At the abs no mention to results or even what are the composition of the 3 alloys are found.
  • Introduction could eventually also include recent examples about the alloy bio applications not only in medicine. Recent papers in this field could be inserted in order to turn the paper with a broader interest.
  • The figures are globally good, but, as possible, could be improved once Metals deserves high quality figures and with rigor in order to avoid lacking of interest for the data. Legends should be also as complete as possible.
  • Globally the conclusions should followed the order of presentation of the paper with partial conclusions first and then global conclusions as well as perspectives.

Author Response

(The authors gave the same response as above.)

Round 2

Reviewer 1 Report

All my comments were taken into consideration and improved in the manuscript. I endorse the paper for publication.

Author Response

Dear Reviewer,

First of all, I would like to thank you for your time dedicating to review my manuscript and for the recommendations

Best regards

Ildiko Peter

Reviewer 2 Report

Dear Authors,

Thank you for considering my remarks.

Author Response

(The authors gave the same response as above.)
